# A Taybi-Linder syndrome-related *RTTN* variant impedes neural rosette formation in human cortical organoids

Justine Guguin[1], Ting-Yu Chen[2], Silvestre Cuinat[1], Alicia Besson[1], Eloïse Bertiaux[3], Lucile Boutaud[4], Nolan Ardito[1], Miren Imaz Murguiondo[5], Sara Cabet[6,7], Virginie Hamel[3], Sophie Thomas[4], Bertrand Pain[8], Patrick Edery[1,9], Audrey Putoux[1,9], Tang K. Tang[2], Sylvie Mazoyer[1], Marion Delous[1]*

1 Université Claude Bernard Lyon 1, CNRS, INSERM, Centre de Recherche en Neurosciences de Lyon CRNL U1028 UMR5292, GENDEV, Bron, France, 2 Institute of Biomedical Sciences, Academia Sinica, Taipei, Taiwan, 3 University of Geneva, Molecular and Cellular biology department, Sciences faculty, Geneva, Switzerland, 4 INSERM UMR 1163, Institut Imagine, Université Paris Cité, Paris, France, 5 Zumarraga Hospital, Pediatric Service, Gipuzkoa, Spain, 6 Service d'imagerie Pédiatrique et Fœtale, Hôpital Femme Mère Enfant, Hospices Civils de Lyon, Bron, France, 7 CNRS, Inserm, Physiopathologie et Génétique du Neurone et du Muscle, Institut NeuroMyoGène, Université de Lyon, Lyon, France, 8 University of Lyon, Université de Lyon 1, INSERM, INRAE, Stem Cell and Brain Research Institute, U1208, USC1361, Bron, France, 9 Unité de génétique clinique et Centre de référence labellisé des Anomalies du Développement Sud-Est, Département de génétique, Hospices Civils de Lyon, Bron, France

* marion.delous@inserm.fr

**Data Availability Statement:** All relevant data are within the manuscript and its Supporting Information files.

## Abstract

Taybi-Linder syndrome (TALS) is a rare autosomal recessive disorder characterized by severe microcephaly with abnormal gyral pattern, severe growth retardation and bone abnormalities. It is caused by pathogenic variants in the *RNU4ATAC* gene. Its transcript, the small nuclear RNA U4atac, is involved in the excision of ~850 minor introns. Here, we report a patient presenting with TALS features but no pathogenic variants were found in *RNU4ATAC*, instead the homozygous *RTTN* c.2953A>G variant was detected by whole-exome sequencing. After deciphering the impact of the variant on the RTTN protein function at centrosome in engineered *RTTN*-depleted RPE1 cells and patient fibroblasts, we analysed neural stem cells (NSC) derived from CRISPR/Cas9-edited induced pluripotent stem cells and revealed major cell cycle and mitotic abnormalities, leading to aneuploidy, cell cycle arrest and cell death. In cortical organoids, we discovered an additional function of RTTN in the self-organisation of NSC into neural rosettes, by observing delayed apico-basal polarization of NSC. Altogether, these defects contributed to a marked delay of rosette formation in *RTTN*-mutated organoids, thus impeding their overall growth and shedding light on mechanisms leading to microcephaly.

## Author summary

Primary microcephaly is defined as a severe reduction of the brain size that occurs prenatally. Variants in about 50 genes have been associated to primary microcephaly, and most

**Funding:** This work was supported by CNRS, Inserm and Université Claude Bernard Lyon 1 through recurrent funding; the Agence Nationale de la Recherche (no. ANR-18CE12-0007; no. ANR-22CE12-0007); the Fondation Jérôme Lejeune and the Fondation pour la recherche sur le Cerveau « Espoir en tête » (confocal microscope). J.G. was supported by the Ministère de l'Enseignement Supérieur et de la Recherche and by the Fondation pour la Recherche Médicale (grant number FDT202304016375), and S.Cu. by a Poste Accueil Inserm 2023. T-Y.C. was supported by a postdoctoral fellowship from Academia Sinica, Taiwan and T.K.T. by the National Science and Technology Council (NSTC 112-2326-B001-010) and Academia Sinica (AS-IA-109-L04), Taiwan. E. B. was supported by an EMBO long-term fellowship (ALTF-284-2019), and V.H. by the Swiss National Foundation (SNSF 310030_205087). S.T. was supported by Agence Nationale de la Recherche (no. ANR-17-CE16-0003-01). The funders had no role in study design, data collection and analysis, decision to publish, or preparation of the manuscript.

**Competing interests:** The authors have declared that no competing interests exist.

of them encode proteins that regulate cell cycle, notably by participating to centrosome biogenesis. Intriguingly, some other genes involved in the process of minor splicing, such as *RNU4ATAC*, are also related to primary microcephaly without clear understanding of the underlying pathophysiological mechanisms. In our previous work, we discovered that alterations of minor splicing result into dysfunction of the centrosome/cilium complex. Here, we further feed this link between minor splicing and centrosome/primary cilium by reporting the particular case of a patient who presents with all features of the rare *RNU4A-TAC*-associated syndrome, called the Taybi-Linder syndrome, and yet, is homozygous for the only recurrent pathogenic variant in the centrosomal *RTTN* gene. Hence, to decipher the underlying cellular mechanisms, we generated unique human neuronal cellular models–iPSC-derived neural stem cells (NSC) and cortical organoids–and unveiled the combination of events that contribute to the depletion of the NSC pool and explain *RTTN*-associated microcephaly. Our work gives thus precious hints for the understanding of the Taybi-Linder syndrome physiopathology.

## Introduction

The Taybi-Linder syndrome (TALS), or MOPD1 (OMIM #210710), is a very rare genetic disease with less than 100 reported cases [1–6] characterized by severe pre- and post-natal growth delay, severe microcephaly with brain abnormalities (abnormal gyral pattern, intracranial cyst, cerebellar vermis hypoplasia and corpus callosum agenesis), intellectual deficiency, bone abnormalities, bulging eyes and dry skin with eczema, as well as a premature death during the first months of life for the most severe cases [7–9]. Since the identification of *RNU4ATAC* in 2011 [10,11], no other gene has been associated with this disease. *RNU4ATAC* is a non-coding gene transcribed into the small nuclear RNA (snRNA) U4atac, a component of the minor spliceosome. This ribonucleic complex excises less than one percent of all human introns [12], and deficiency in its function leads in most cases to minor intron retention that impairs the correct maturation of about 750 minor-intron containing transcripts [2,13–17].

Here, we report on a patient with clinical features of TALS but no variants in *RNU4ATAC*. Whole exome sequencing rather revealed she has the homozygous pathogenic missense variant NM_173630: *RTTN* c.2953A>G, p.Arg985Gly, which also alters splicing. This gene has already been implicated in human disease, notably "microcephaly, short stature and polymicrogyria with or without seizure" (MSSP, OMIM #614833). Up to now, 39 patients from 23 families have been reported with bi-allelic pathogenic variants in *RTTN* [18–33] of which 4 patients, from 3 families, have the c.2953A>G variant. Most of these patients exhibit common features, at variable degrees of severity, including microcephaly with gyral pattern abnormalities, sloping forehead, intellectual deficiency, and pre- and post-natal growth delay. A few patients only were deceased at the time of publication, with one third of them having reached teenage- or adulthood.

*RTTN* encodes the large centrosomal protein Rotatin (2,226 amino-acids), and was initially discovered by a forward genetic screen in the mouse model, with the particular phenotype of *Rttn* mutant embryos failing to undergo the axial rotation during embryogenesis [34]. Further investigations demonstrated that Rotatin is essential for procentriole elongation [35], correct cell cycle progression and mitosis [29,36], and regulation of primary cilium length [18,29]. Alterations of some of these processes have been shown to contribute to the development of microcephaly [37].

Intrigued by the TALS-like phenotype of the patient with this *RTTN* variant, we thought to investigate the associated pathophysiological mechanisms leading to microcephaly. For that, we used patient fibroblasts, engineered RPE1 cells and CRISPR/Cas9-edited induced pluripotent stem cells (iPSC) that were differentiated into neural stem cells (NSC) or cortical organoids (CO). We carried out a thorough analysis of the impact of the *RTTN* isoforms resulting from the mutated allele identified in the patient and conducted cellular studies of the centrosome/primary cilium complex and their related functions. Overall, our results establish how *RTTN* c.2953A>G variant affects the integrity of the neural stem cell pool, thus leading to improper neuronal cell mass in patients.

## Results

### Identification of *RTTN* variant in a patient with TALS-like features

Blood sample from a child suspected to have Taybi-Linder syndrome was addressed to us for genetic diagnosis. The patient is the only daughter of healthy Moroccan first-cousin parents without any notable medical history reported in the family (Fig 1A). The pregnancy was marked by a severe intrauterine growth retardation with brain abnormalities. The baby girl was born at 39 weeks of gestation with a weight of 1695 g (-4 SD; <0.01 percentile), a height of 43 cm (-4 SD; 0.04th percentile) and a head circumference (HC) of 22.5 cm (-9 SD; <0.01 percentile). Radiographic findings revealed a global ossification delay, with a bone age of 10 months for a chronological age of 15 months, with a prominent occiput, 11 pairs of ribs, but no skeletal dysplasia. In her first months of life, she developed severe, diffuse eczema with alopecia. She showed a moderate global developmental delay, and her growth retardation worsened, while maintaining harmonious morphology and proportions: at 20 months, she weighed 6600g (-4 SD; <1st percentile), measured 68 cm (-4.5 SD; <1st percentile) and had a HC of 34.5 cm (-10 SD; <1st percentile). At 9 years old, HC was 39 cm (-10.5 SD; <1st percentile). At 13 years old, she weighed 35.1 kg (-1.5 SD; 10th percentile) and her height was 131 cm (-4 SD; <1st percentile). She had a low frontotemporal hair implantation, an extremely sloping forehead, a discrete convergent strabismus, and retrognathism. Brain MRI performed at 20 months of age showed microcephaly with short but complete corpus callosum (arrowhead, Fig 1B), delayed myelination and supratentorial pachygyria (red arrows, Fig 1C). The cerebellum and brainstem were normal (Fig 1B). She is currently aged 17 and has a neurological follow-up for cognitive impairment and epilepsy treated with valproate, clobazam and olanzapine.

Because of the clinical suspicion of TALS, Sanger sequencing of *RNU4ATAC* was first performed, but no variant was found. Then, whole exome sequencing was done and the homozygous variant NM_173630: *RTTN* c.2953A>G was identified (ClinVar #977819, reported as pathogenic), in a gene that does not contain any minor intron. Interestingly, this case is not the first of its kind. When we reviewed the literature on cases harboring bi-allelic variants in *RTTN*, two families reported by Grandone *et al.* [20] and by Shamseldin *et al.* [19] particularly caught our attention. In the former report, the siblings, a boy and a girl, who were also homozygous for the *RTTN* c.2953A>G variant, were not formally diagnosed with TALS but they presented all the typical features of it: severe growth retardation, severe microcephaly (with lissencephaly), eczema and developmental delay. In the latter, two siblings, negative for *RNU4ATAC* but having compound heterozygous missense variants in *RTTN*, were clinically diagnosed with MOPD1 as they were presenting severe pre- and post-natal growth delay, severe microcephaly (with cerebellar hypoplasia, dysgenesis/agenesis of corpus callosum, lissencephaly or reduced sulcation, ventricular abnormalities), joint contractures and the typical facial dysmorphism (prominent eyes, sloping forehead, micrognathia). The two baby boys

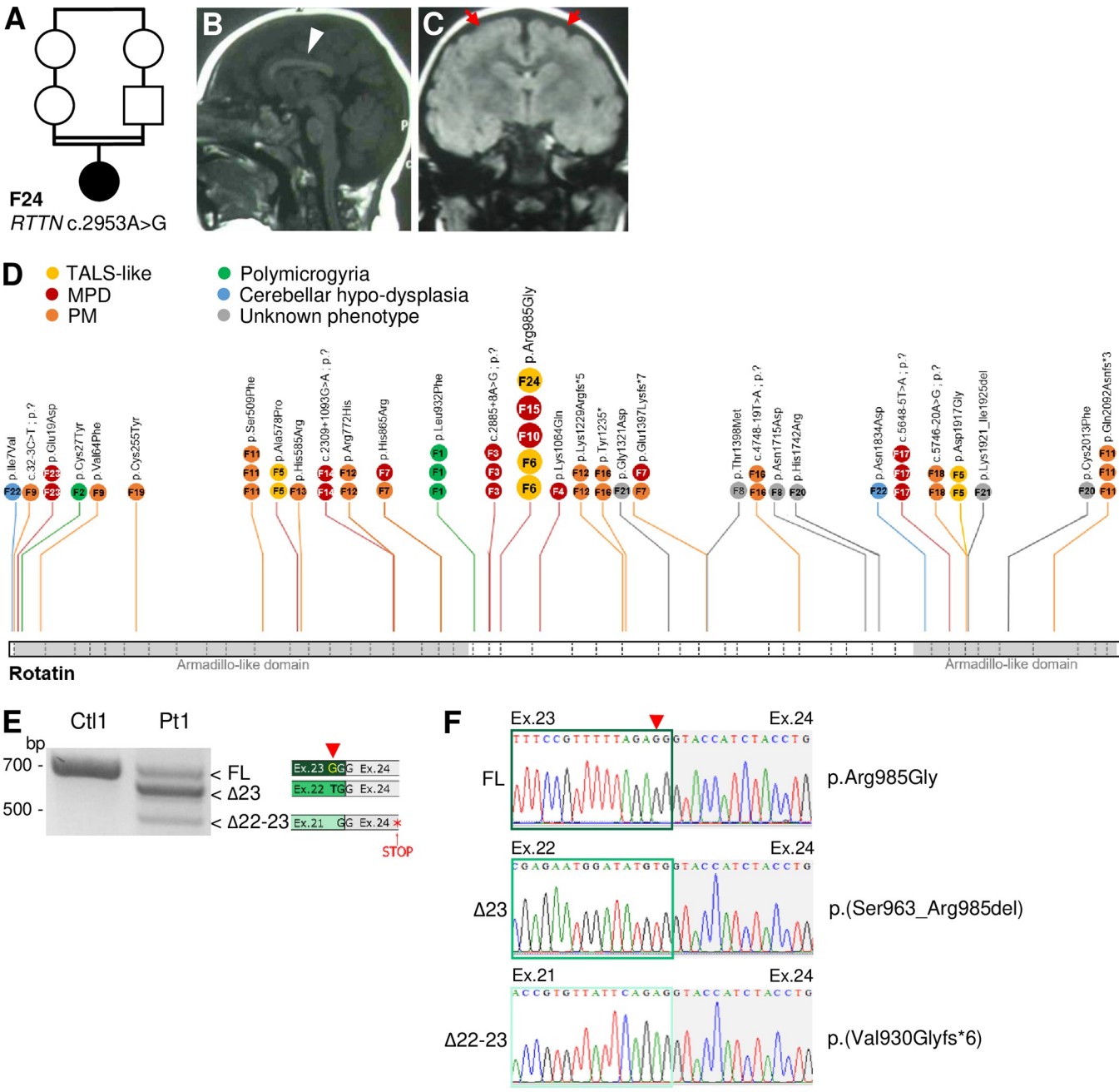

**Fig 1. *RTTN* c.2953A>G variant is associated to TALS-like phenotype and causes splicing alteration of exon 23. (A)** Pedigree of the TALS-like patient's family (F24 in S1 Table). **(B, C)** Brain MR images on midsagittal T1WI (B) and coronal T2-FLAIR (C) sequences, performed at 17 months, showing a short but complete corpus callosum (arrowhead on B) and a supra-tentorial diffuse pachygyria (red arrows on C). Delayed myelination for age was also noted on axial sequences. **(D)** Schematic representation of Rotatin protein showing the functional domains (Armadillo-like domains) and the position of variants in light of the patients' phenotype. All variants, but c.2953A>G, p.Arg985Gly, seen in 4 families, were identified in one family (F) only, numbered according to S1 Table. Circles indicate the number of patients per family. Dashed lines on the protein structure indicate exon-exon junctions. **(E)** RT-PCR analysis of exon 21 to exon 25 fragment in fibroblasts. Three splicing events are observed: a full-length (FL) form that includes the nucleotide change (red arrowhead), a form depleted of exon 23 (Δ23) and a form depleted of both exons 22 and 23 (Δ22–23) that leads to premature stop codons from exon 24. **(F)** Sanger sequencing of the patient PCR amplicons seen in D. MPD, Microcephalic Primordial Dwarfism; PM, Primary Microcephaly.

died before 3 months of age. Of note, all the other *RTTN* published cases present with quite heterogeneous clinical presentations, distinct from the TALS phenotype (Fig 1D and S1 Table).

Since the pathophysiological mechanisms by which damaging *RTTN* variants impact brain development are poorly known, we investigated *RTTN* c.2953A>G further through the study of various cellular models, which provided us with insights in the molecular mechanisms of disease associated with Taybi-Linder syndrome.

### *RTTN* c.2953A>G variant leads to in-frame exon 23 skipping that is responsible for cell division defects

As previously described [20,24,29], *RTTN* c.2953A>G variant, located at the penultimate nucleotide of exon 23, alters the splicing donor site sequence of intron 23. We first confirmed by RT-PCR, RT-qPCR and RNA sequencing that c.2953A>G induces partial skipping of exon 23 in patient fibroblasts (Figs 1E, 1F, S1A and S1B). Indeed, the variant results in the expression of three different RNA isoforms: the full length (FL) isoform, which encodes the amino-acid change p.Arg985Gly (RG); the most prevalent isoform, resulting from the skipping of the in-frame exon 23, which encodes a Rotatin protein depleted of 23 amino-acids p.(Ser963_Arg985del) (Δ23); and the third isoform, resulting from the skipping of both exons 22 and 23 and which contains premature stop codons, p.(Val930Glyfs*6) (Δ22–23). This latter isoform is partially degraded by nonsense-mediated mRNA decay (S1B Fig) and is likely to give rise to small amounts of a nonfunctional truncated protein that lacks two third of its full length. Globally, the alteration of *RTTN* pre-mRNA splicing does not alter its total level of expression (S1C Fig). To investigate the consequences of *RTTN* c.2953A>G at the protein level, we performed ultrastructural expansion microscopy (U-ExM), allowing precise nanoscale mapping of proteins. We first confirmed in control cells that Rotatin is located at the proximal end of centrioles, lining the internal microtubule wall in both mature and pro-centrioles, and has a ring-like localization from top-viewed centrioles (Figs 2A and S1D). Rotatin is also present at the basal body, without any signal detected in the axoneme of primary cilia (S1D Fig). In patient cells, we observed a modest reduction in Rotatin quantity at the centrioles (Fig 2A and 2B), suggesting a defective targeting or recruitment of mutated Rotatin, which was accompanied by a slight decrease of centriole length (Fig 2C). Of note, and contrary to what was previously published [29], we observed no alteration of cell cycle progression and mitosis events in patient fibroblasts (S2A–S2D Fig).

To evaluate the respective effect of the missense p.Arg985Gly and the Δ23 variants, we infected the previously described double knock-out *p53*$^{-/-}$*;RTTN*$^{-/-}$ (*RTTN*-dKO) RPE1 cells [36] with each of the two doxycycline-inducible GFP-tagged RTTN isoforms. We first verified that they were correctly expressed by immunofluorescence and western blot (Figs 2D, S3A and S3B). Contrary to WT and RG Rotatin proteins which both localize to mature and pro-centrioles, we observed that the Δ23 Rotatin abundance at centrioles was highly decreased (Fig 2D and 2E). As a consequence, loading of downstream centriolar proteins such as POC1B [35] was altered in presence of Δ23 Rotatin (S3C and S3D Fig). Following the decreased localization of both Rotatin and POC1B to centrioles, we observed a shortening of centrioles in *RTTN*-dKO RPE1 cells expressing Δ23 Rotatin (Fig 2F and 2G). We also observed a slight G2/M-phase retention and aneuploidy (Fig 2H and 2I), as well as an increase of abnormal mitotic events (Fig 2J and 2K), in *RTTN*-dKO cells induced to express Δ23 Rotatin compared to the WT form [36].

Hence, taken together, these results show that the Δ23 Rotatin protein has the most deleterious effect, and that the loss of 23 amino-acids has not only a deleterious impact on the targeting or recruitment of Rotatin to centrioles, but that it also alters its function in recruiting downstream effectors of centriole maturation. These alterations may explain the defective division of cells.

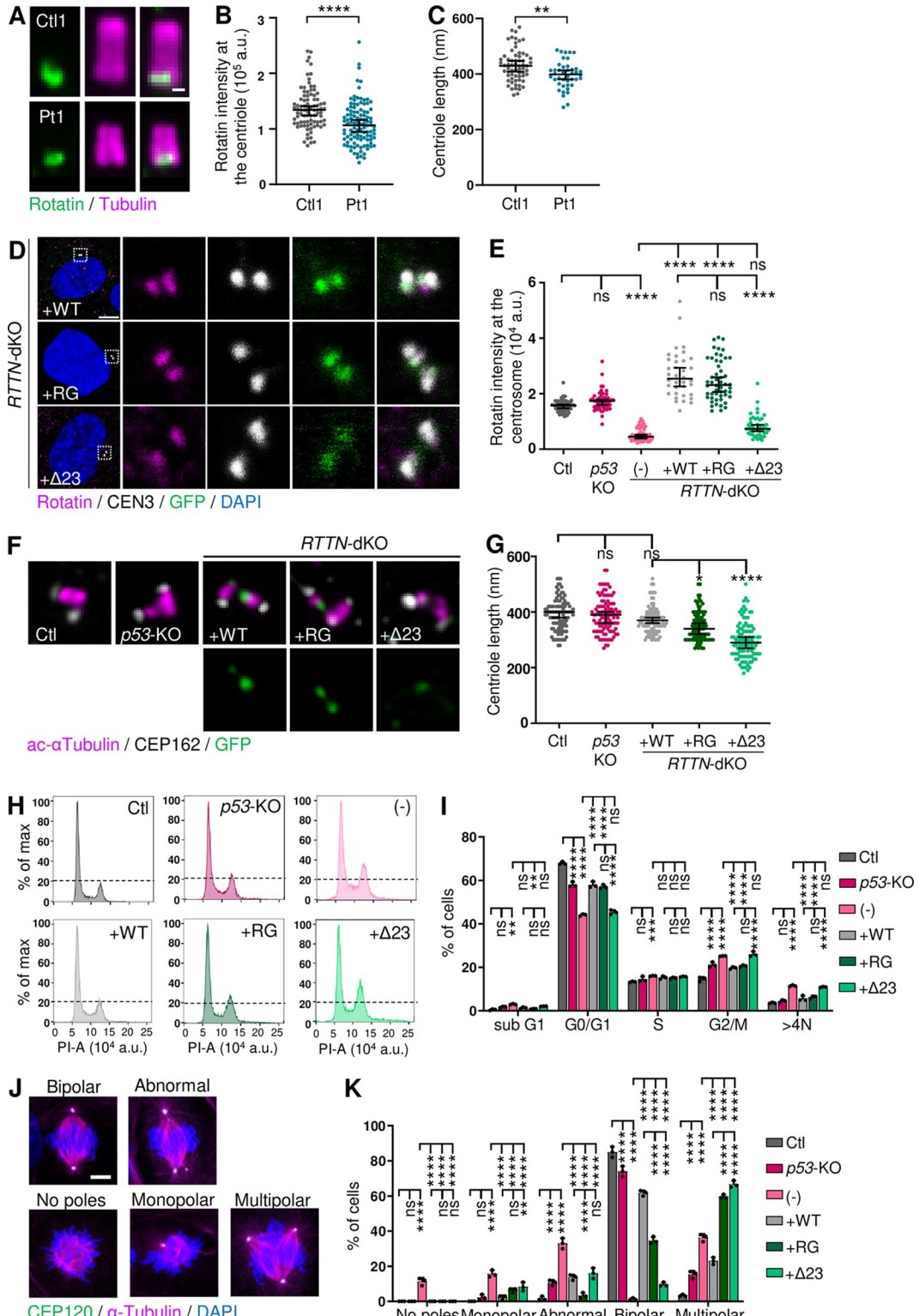

**Fig 2.** *RTTN* **c.2953A>G variant alters Rotatin localization at centriole, cell cycle progression and mitosis in *RTTN*-dKO RPE1 cellular model.** Experiments were performed in control (Ctl1) and patient (Pt1) fibroblasts (A-C), or in control, *p53*-KO and *RTTN*-dKO RPE1 cells induced to express wild-type (WT) or mutated (RG, Δ23) RTTN-GFP proteins (D-K). **(A)** Representative confocal images of Rotatin (green) in expanded centrioles, labeled with tubulin (magenta), in fibroblasts. **(B, C)** Quantification of Rotatin intensity at the centrioles (B) and of length of centrioles (C), such as shown in A. Graphs

show the median ± 95% CI from three independent experiments (n>85 (B) or n>40 (C) centrioles). **(D)** Representative confocal images of Rotatin (magenta) localisation at the centrosome in *RTTN*-dKO RPE1 cellular model. Centrin3 (CEN3, grey) labels centrosomes. **(E)** Quantification of Rotatin intensity, relative to CEN3, at the centrioles, such as shown in D and in S2A Fig. Graph shows the median ± 95% CI from three independent experiments (n>35 centrosomes). **(F)** Representative super-resolution images of centrioles in *RTTN*-dKO RPE1 cellular model. Acetylated α-Tubulin (magenta) labels centriole structure and CEP162 its distal end. **(G)** Quantification of length of centrioles, such as seen in F. Graph shows the median ± 95% CI from three independent experiments (n≥100 centrioles). **(H)** Flow cytometric cell cycle analysis histograms of *RTTN*-dKO RPE1 cellular model. The dotted line represents the top of G2/M peak in Ctl cells for reference. **(I)** Quantification of the percentage of cells in each cell cycle phase, such as seen in H. Graph shows the mean ± SD of three independent experiments. **(J)** Representative confocal images of metaphases seen in *RTTN*-dKO RPE1 cellular model. Acetylated α-Tubulin labels mitotic spindles and CEP120 centrosomes. **(K)** Quantification of the proportion of cells with the indicated mitotic phenotypes. Graph shows the mean ± SD of three independent experiments (n ≥ 100 cells per experiment). ns, not significant; *p-value<0.05; **p-value<0.01; ***p-value<0.001; ****p-value<0.0001 following Mann-Whitney test (B), t-test (C), Kruskal-Wallis with Dunn's Multiple Comparison test (E, G) or two-way ANOVA with Tukey's correction (I, K). Scale bars: 100 nm (A), 5 μm (D). DAPI stains DNA. a.u. arbitrary units, PI propidium iodide.

### *RTTN* c.2953A>G variant disrupts cilium formation and function

As aforementioned, Rotatin localizes to the base of primary cilia in quiescent cells (S1D Fig) and previous reports showed that primary cilium length was significantly reduced in some patient fibroblasts [18,23,29]. Hence, we investigated the formation, disassembly and function of primary cilia as a consequence of c.2953A>G.

First, and contrary to what has been previously reported for other *RTTN* variants [18,23,29], we did not observe any alterations of proportion of ciliated cells or cilium length in the patient fibroblasts (Fig 3A–3C). However, in *RTTN*-dKO RPE1 cells, we did observe a drastic loss of primary cilium formation, that was partially rescued by the WT and RG Rotatin forms (both count and length) but not by the Δ23 variant (S4A–S4C Fig). We thus concluded that Rotatin is essential for cilium formation–as it was shown by RNA interference [18]–but that the *RTTN* c.2953G>A variant in patient fibroblasts may produce enough RG protein to allow cilium formation.

Next, we performed a dynamic analysis of cilium disassembly in patient fibroblasts. After promoting cilium formation for 48 hours by serum starvation, ciliary disassembly was induced upon serum addition. While the percentage of ciliary cells was unchanged in both control and patient cells, we observed a decrease of cilium length in control cells from T = 3 hours following serum addition that was less pronounced in patient cells (Fig 3D–3F), suggesting that loss of function of Rotatin may lead to a delayed cell cycle re-entry, which is an important parameter in highly proliferative cells such as neural stem cells, and that has been shown to contribute to microcephaly [38–42].

Finally, we investigated primary cilium function by analyzing two ciliary signaling pathways. The first one is the Sonic Hedgehog (Shh) pathway, in which the binding of the ligand Shh to its ciliary receptor Ptch1 alleviates the inhibitory signal that maintains Smoothened (Smo) out of the cilium. Upon Shh activation, Smo translocates into the cilium and activates the processing machinery of Gli transcription factors. Once cleaved, these factors shuttle into the nucleus to control Shh target gene expression (including *GLI1*). Upon treatment with the Smo agonist SAG to activate the Shh pathway, we observed an upregulation of *GLI1* expression in control cells, which is slightly higher in patient fibroblasts (albeit not significantly) (Fig 3G). As a second ciliary signaling pathway, we investigated the ciliary localization of the adenylate cyclase III (AC3) enzyme that locally generates cAMP from ATP and participates, among others, to the control of cilium length and Shh pathway [43]. We observed a significant decrease in AC3 ciliary localization in patient fibroblasts (Fig 3H and 3I), thus suggesting that Rotatin is required for the proper ciliary localisation of AC3 and other signaling molecules to ensure cilium function.

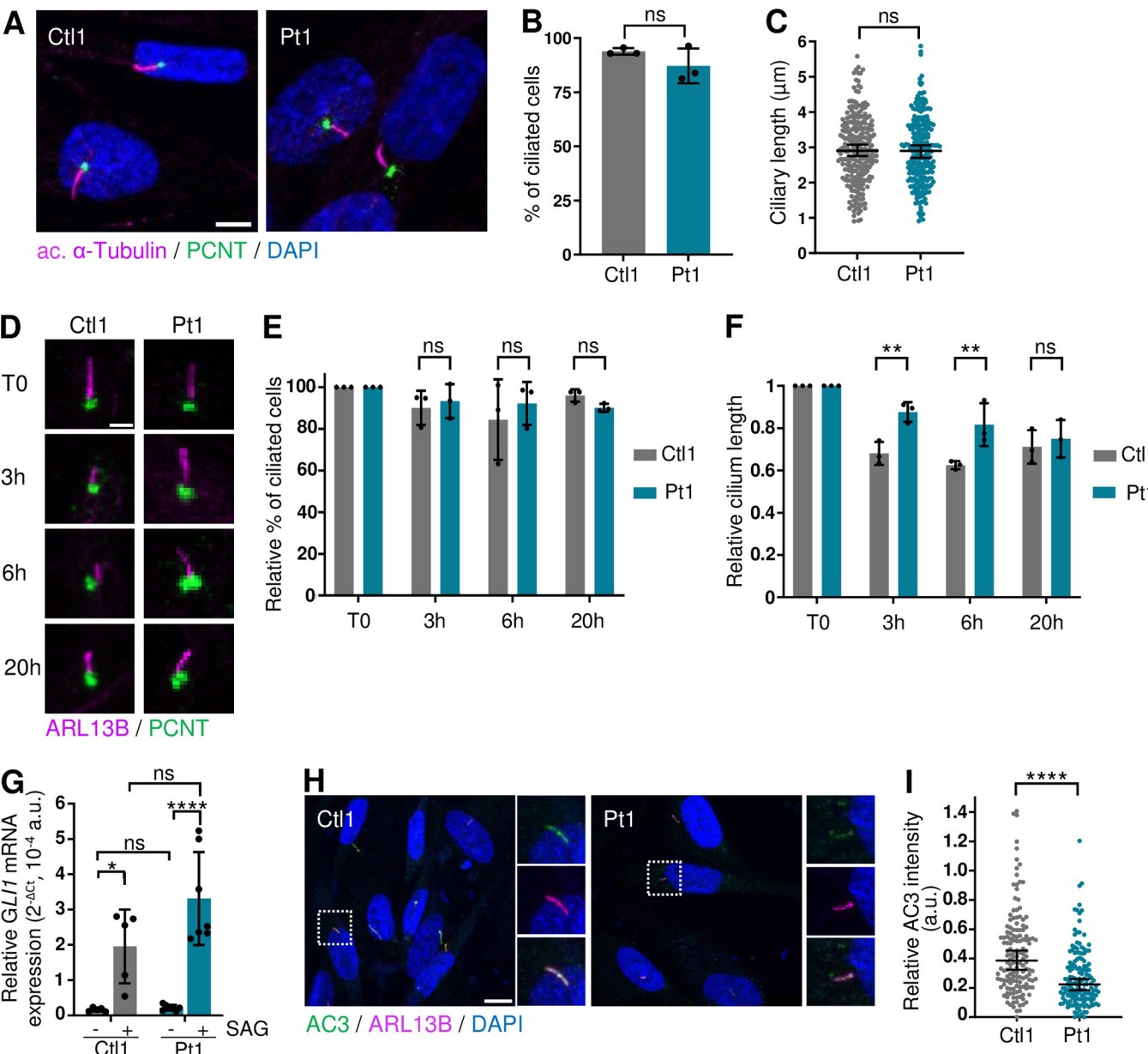

**Fig 3. *RTTN* c.2953A>G variant alters ciliary function in patient fibroblasts.** All experiments were performed in control (Ctl1) and patient (Pt1) fibroblasts. **(A)** Representative confocal images of primary cilium (acetylated α-Tubulin, magenta) and centrosome (PCNT, green) in fibroblasts. **(B, C)** Quantification of percentage of ciliated cells (B) and of length of cilia (C) observed in A. Graphs show the mean ± SD (B) or median ± 95% CI (C) of three independent experiments (n = 250 cells). **(D)** Representative confocal images of disassembly of primary cilium (ARL13B, magenta) in fibroblasts, at serum starvation (T0) or at T = 3, 6 or 20 hours after serum addition in the culture medium. PCNT (green) labels centrosome. **(E, F)** Quantification relative to T0 of percentage of ciliated cells (E) and of length of cilia (F) observed in D. Graphs show the mean ± SD of three independent experiments (n>100 cells). **(G)** RT-qPCR analysis of *GLI1* relative expression in fibroblasts not treated or treated with SAG. *ACTB* was used as house-keeping gene. Graph shows the mean ± SD of five independent experiments. **(H)** Representative confocal images of adenylate cyclase 3 (AC3, green) in primary cilium (ARL13B, magenta) in fibroblasts. **(I)** Quantification of ciliary AC3 staining intensity relative to ARL13B. Graph shows the median ± 95% CI of three independent experiments (n = 150 cells). ns, not significant; *p-value<0.05; **p-value<0.01; ***p-value<0.001; ****p-value<0.0001 following Mann-Whitney test (B, C, I), two-way ANOVA with Bonferroni's correction (E, F), two-way ANOVA with Tukey's correction for multiple tests (G). Scale bars: 5 μm (A), 2 μm (D), 10 μm (H). DAPI labels nuclei. a.u. arbitrary units.

Altogether, these results show that Rotatin is required for both cilium formation and function, and that Δ23 Rotatin is once again the most deficient form.

### *RTTN* c.2953A>G variant leads to cell cycle defects, abnormal mitotic events and increased cell death in iPSC-derived neural stem cells

To be able to better investigate the pathophysiological mechanisms involved in the microcephaly and brain abnormalities seen in the patient, we generated CRISPR/Cas9-edited induced pluripotent stem cells (iPSC) (S1 Methods and S5A–S5D Fig). In iPSC, the c.2953A>G variant had similar impact on *RTTN* pre-mRNA splicing as that seen in fibroblasts, except that the Δ23 isoform was more abundant (S5E and S5F Fig); as in fibroblasts, the levels of mRNA expression were unchanged (S5G Fig). A comprehensive analysis of CRISPR/Cas9-edited iPSC phenotypes showed no significant differences between WT and *RTTN* mutated (KI) clones regarding cell cycle progression (S6A and S6B Fig), mitotic events (S6C and S6D Fig), percentage of ciliated cells and cilium length (S6E–S6G Fig).

The iPSC were then differentiated into neural stem cells (NSC) (S7A–S7C Fig). In this cell type, at 25 days *in vitro* (DIV25), the proportion of the three mRNA isoforms resulting from the presence of c.2953A>G was as seen in fibroblasts (S7D–S7F Fig). No strong alteration of centriole length was detected in KI NSC (S7G and S7H Fig), neither significant abnormalities regarding the percentage of ciliated cells, cilium length or function (as far as AC3 ciliary localization is concerned) (S8A–S8E Fig); however, we observed by flow cytometry a higher proportion of KI NSC with aneuploidy and with retention in G2/M phase at the expense of phase G1 compared to WT NSC (Fig 4A and 4B). Careful analysis of mitosis events by immunofluorescence allowed the detection of various defects of the mitotic spindles in KI NSC: whereas in WT NSC, 90% of mitosis were normal with 2 spindle poles at the metaphase, more than 40% of mitosis in KI NSC showed anomalies, such as no poles, one pole, asymmetric poles or multiple poles (Fig 4C and 4D), thus confirming the observation seen by flow cytometry. To determine the fate of the cells with defective mitosis, we investigated markers of cell arrest and apoptosis. We observed a higher number of double-positive p53+; p21+ cells in KI NSC compared to controls, as well as an increase of cleaved-Caspase 3 staining (Fig 4E–4H), indicating that a higher proportion of KI NSC underwent cell cycle arrest and/or apoptosis. Then, to further explore mitotic events, we investigated spindle orientation by determining the angle formed by the two spindle poles [36]. Whereas in both WT NSC lines, the two centrosomes were mostly aligned within a plane parallel to the support (between 5 and 10˚), they tended to be slightly misaligned in KI NSC lines (15–30˚) (Fig 4I and 4J). This result suggests that a higher proportion of KI NSC could be prone to undergo asymmetric divisions and thus to prematurely shift to neurogenic cell divisions, which, as a result, would exhaust the progenitor pool and lead to microcephaly [39,44,45].

### *RTTN* c.2953A>G-mutated cortical organoids display growth defects with delayed neural rosette formation and increased cell death

We generated matrix-free cortical organoids (CO) derived from CRISPR/Cas9-edited iPSC (S9A–S9C Fig), in order to mimic, in three dimensions, the early stages of brain development. Indeed, the organization of NSC into neural rosettes resembles the developmental structures of neural tubes. First, to check if the splicing outcomes of *RTTN* pre-mRNA with c.2953A>G variant was modified in the context of an increased cellular complexity, we analyzed the RNA isoforms in DIV35 CO. We found the same three isoforms as before, in almost the same proportions as in fibroblasts and NSC, except that the FL isoform was less abundant (S9D and S9E Fig). Variations in total *RTTN* mRNA expression did not correlate with the genotypes (S9F Fig).

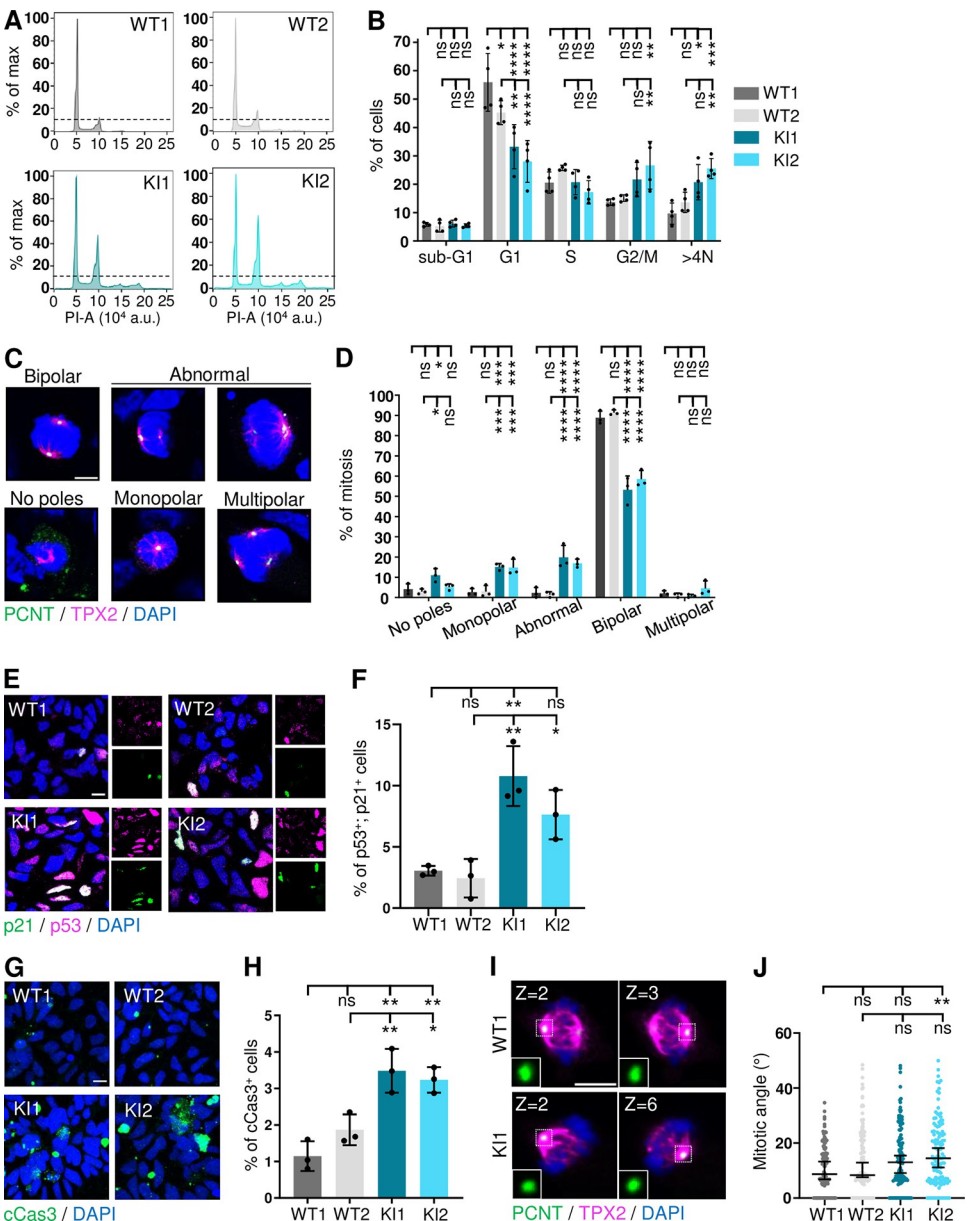

**Fig 4.** *RTTN* **c.2953A>G variant alters mitosis and cell cycle progression in neural stem cells.** All experiments were performed in control (WT) and *RTTN*-mutated (KI) NSC. **(A)** Flow cytometric cell cycle analysis histograms in NSC. The dotted line marks the maximal value of G2/M peak in WT1 cells for reference. **(B)** Quantification of the percentage of cells in each cell cycle phase. Graph shows the mean ± SD of three independent experiments. **(C)** Representative confocal images of metaphases observed in NSC. PCNT (green) labels the centrosomes, TPX2 (magenta) the mitotic spindle. **(D)** Quantification of the proportion of cells with the indicated mitotic phenotypes. Graph shows the mean ± SD of three independent experiments (n>100 mitosis per experiment). **(E)** Representative confocal images of cells stained with p53 (magenta) and p21 (green), markers of cell arrest in NSC. **(F)** Quantification of the percentage of p53; p21-double positive cells, such as seen in E. Graph shows the mean ± SD of three independent experiments (n>800 cells per experiment). **(G)** Representative confocal images of apoptotic cells (cleaved Caspase-3, green) in NSC. **(H)** Quantification of the percentage of cCas3-positive cells, such as seen in G. Graph shows the mean ± SD of three independent experiments (n>800 cells per experiment). **(I)** Representative confocal images of metaphases in NSC. PCNT (green) labels the centrosomes, TPX2 (magenta) the mitotic spindle. In insets are shown each of the two spindle poles located in the indicated Z-plane. **(J)** Quantification of the mitotic angle based on the distance and the height between both spindle poles. Graph shows the median ± 95% CI from three independent experiments. ns, not significant; *p-value<0.05; **p-value<0.01; ***p-value<0.001; ****p-value<0.0001 following two-way (B, D) or one-way (F, H) ANOVA with Tukey's correction, or Kruskal-Wallis with Dunn's correction (J). Scale bars: 5 µm (C, I), 10 µm (E, G). DAPI labels nuclei. a.u., arbitrary units.

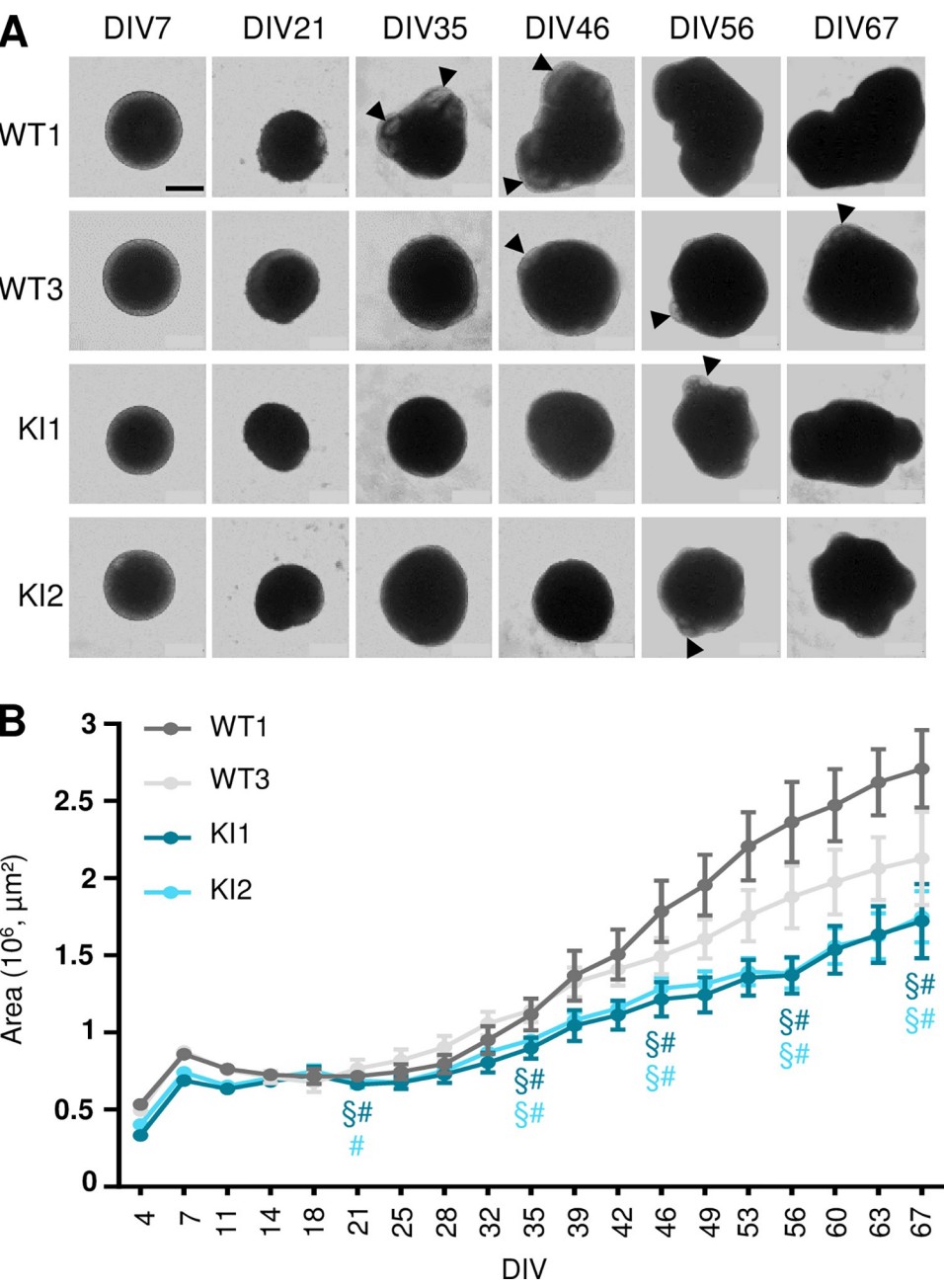

**Fig 5. *RTTN* c.2953A>G variant alters cortical organoid growth.** All experiments were performed in wild-type (WT) and *RTTN*-mutated (KI) cortical organoids (CO). **(A)** Representative optical images of CO at different time points, from DIV7 to DIV67. Arrowheads indicate neural rosettes seen in transparency. **(B)** Growth curve of CO such as seen in A. Graph shows the mean ± SD of 20 CO areas at each time point. Statistical analyses are shown for DIV21, DIV35, DIV46, DIV56 and DIV67 only. § and # indicate a p-value<0.001 when compared to WT1 and WT3 respectively, following two-way ANOVA with Tukey's multiple correction. Scale bar: 500 nm. DIV, days in vitro.

We then monitored CO growth by measuring their size twice a week, over a period of culture of 67 days. We observed that starting at DIV21, KI CO exhibited a slower growth rate compared to WT CO, being more than half smaller than WT1 at DIV67 (Fig 5A and 5B). In addition to alteration of CO size, we noticed that KI CO morphology was altered: while neural

rosettes were appearing at the periphery in WT CO as early as DIV35, these structures were seen in KI CO from DIV56 onwards only (Fig 5A, arrowheads).

Next, to have a better understanding of neural rosette formation in the context of the presence of c.2953A>G, we performed immunostaining on CO cryosections, using SOX2 to label NSC and N-cadherin to label their apical adherent junctions facing the rosette lumen. By quantifying their number, we found that rosettes in KI CO were overall fewer compared to both WT CO, notably at DIV46 and DIV56 (Fig 6A and 6B). At these same stages of CO

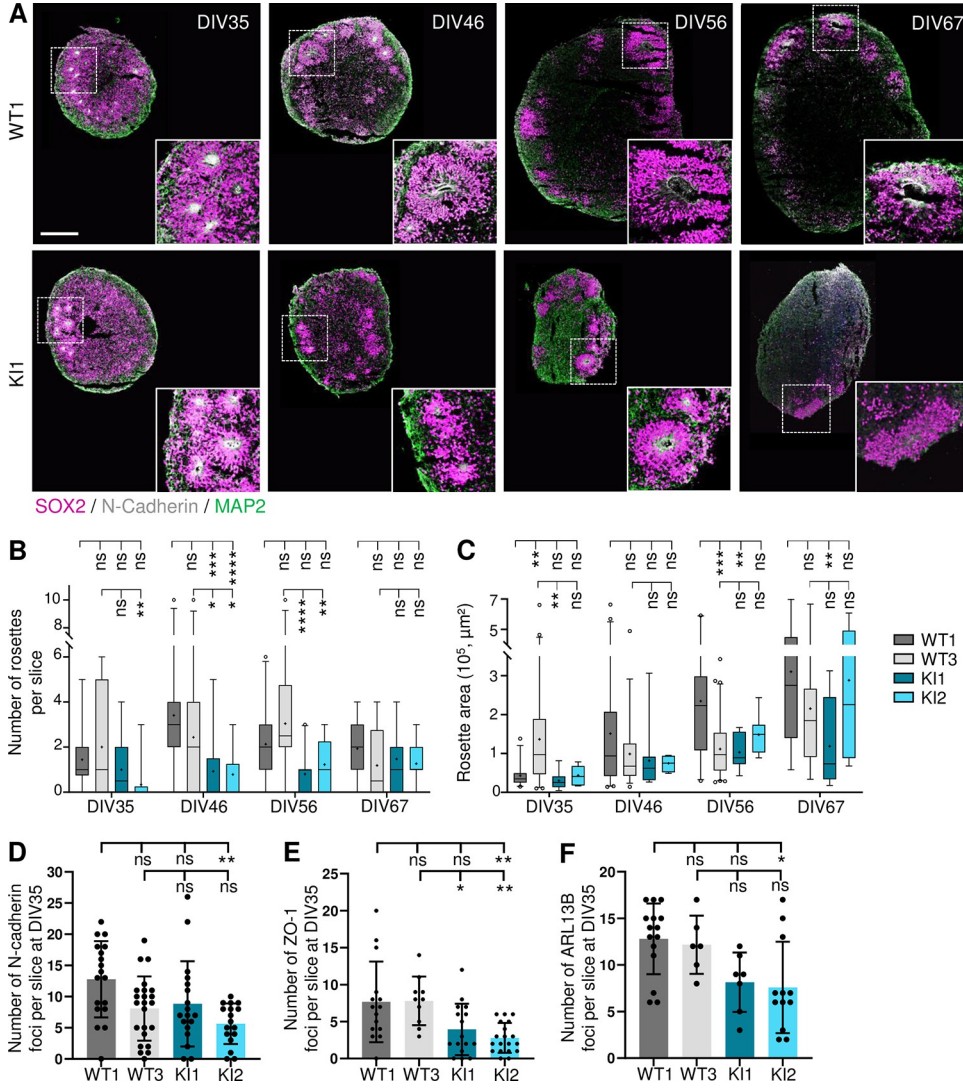

**Fig 6. *RTTN* c.2953A>G variant impedes the formation of neural rosettes in the cortical organoids.** All experiments were performed in wild-type (WT) and *RTTN*-mutated (KI) cortical organoids (CO). **(A)** Representative confocal images of neural rosettes in CO sections at DIV35, DIV46, DIV56 and DIV67. SOX2 (magenta) labels neural stem progenitors (NSC), N-cadherin (grey) the rosette lumen, and MAP2 (green) neurons. **(B-C)** Quantification observed overtime of the number (B) and area (C) of the rosettes, such as seen in A. Box-and-whisker plots display, in the box, the median (middle bar), the mean (cross) and the 25th-75th percentiles, and in whiskers, the 5th to 95th percentile of values from 4 different organoids (n = 6 to 20 slices in total). **(D-F)** Quantification of the number of N-cadherin (D), ZO1 (E) and ciliary (F) foci observed in CO at DIV35. N-Cadherin is a marker of the adherent junctions (as seen in A), ZO1 the tight junctions, and ARL13B the cilium. Graphs show the mean ± SD from 4 different CO (n = 6 to 20 slices total). ns, non-significant; *p-value<0.05; **p-value<0.01; ***p-value<0.001; ****p-value<0.0001 following two-way ANOVA with Tukey's correction (B, C) or Kruskal-Wallis with Dunn's multiple comparisons test (D-F). Scale bar: 250 μm. DIV, days in vitro.

development, neural rosettes seemed also smaller and less well-organised in KI CO compared to WT1 (Fig 6A and 6C). Indeed, SOX2+ cells appeared less elongated and less stratified in the rosette width, as indicated by the overall decreased thickness of neural rosettes at DIV46 and DIV56, while rosette lumen area remained unchanged (S9G and S9H Fig). We thus concluded from these experiments, and despite the phenotypic variability between WT clones, that the presence of *RTTN* c.2953A>G leads to the delayed formation of neural rosettes. In accordance with that, we noticed that the number of N-cadherin-positive foci, a mark of NSC polarity within neural rosettes, tended to be lower in KI CO at DIV35 compared to controls (Fig 6A and 6D). This finding was further supported by the study of two other NSC apical markers, i.e. ZO-1 (tight junctions) and ARL13B (cilia), which showed the same decreased tendency in KI CO compared to controls (Fig 6E and 6F). Hence, these results are in favor of the hypothesis that delayed formation of neural rosettes in KI CO may be due to alteration of NSC polarization, which remains to be investigated.

Finally, to verify if a defect in cell proliferation could also be involved in the decreased rosette number observed in *RTTN*-mutated CO, we performed immunostainings with Ki67 (Fig 7A). We observed a decrease of Ki67 staining in both KI CO compared to WT1 (Fig 7A and 7B), which was concomitant with a decreased number of SOX2+ cells (Fig 7A and 7C), thus suggesting an overall lower number of cycling NSC in *RTTN*-mutated CO. We further analyzed NSC divisions that occur at the center of neural rosettes, and performed immunostainings to label dividing NSC with TPX2, a marker of spindles, and phospho-Vimentin (Fig 7E). We observed that the dynamic of mitotic event occurrence per rosette was slower in KI compared to control CO. In control CO, mitosis were mostly seen at DIV35 and DIV46, whereas in KI CO, they were more frequent at DIV67 (Fig 7F). Finally, by analyzing cell apoptosis, we observed an increase of cleaved caspase-3 staining in KI neural rosettes compared to controls, especially at DIV56 (Fig 7D).

Hence, altogether, these data suggest that the alteration of several cellular processes, including NSC polarization, division and survival, contributes to the delay of rosette maturation in *RTTN*-mutated CO.

## Discussion

In this article, we investigated the pathophysiological mechanisms underlying the pathogenicity of the *RTTN* c.2953A>G variant identified in a patient with a Taybi-Linder-like syndrome, in fibroblasts derived from the patient, and for the first time, in human iPSC-derived neuronal models. We showed that this variant mostly exerts its pathogenicity by inducing the skipping of in-frame exon 23 that encodes 23 amino-acids, and that it has a pleiotropic impact on diverse cellular processes occurring concomitantly or in cascade, including alterations of cell organization/polarity, division and survival of neural progenitors. All these alterations contribute to NSC pool reduction, thus leading to the microcephaly seen in the patient we describe here and probably, in most patients having *RTTN* variants.

The *RTTN* c.2953A>G variant is present in a total of 5 out of 40 reported cases (in 4 out of 24 families), who all originate from North Africa, thus suggesting a founder effect (Fig 1D, and S1 Table) [20,24,29]. It has been reported in the gnomAD database at the heterozygous state in 4 individuals only (out of 725,468 sequenced), all of them not unambiguously clustering with the major populations in a principal component analysis. It is noteworthy that it is the only recurrent *RTTN* variant, all the other being private variants (Fig 1D). Whereas in previous reports it was suspected that patient symptoms resulted from the combined effects of the expression, at the protein level, of both the missense p.Arg985Gly and the p.Ser963_Arg985del (Δ23) Rotatin forms [20,24,29], here we show that the missense p.Arg985Gly has low

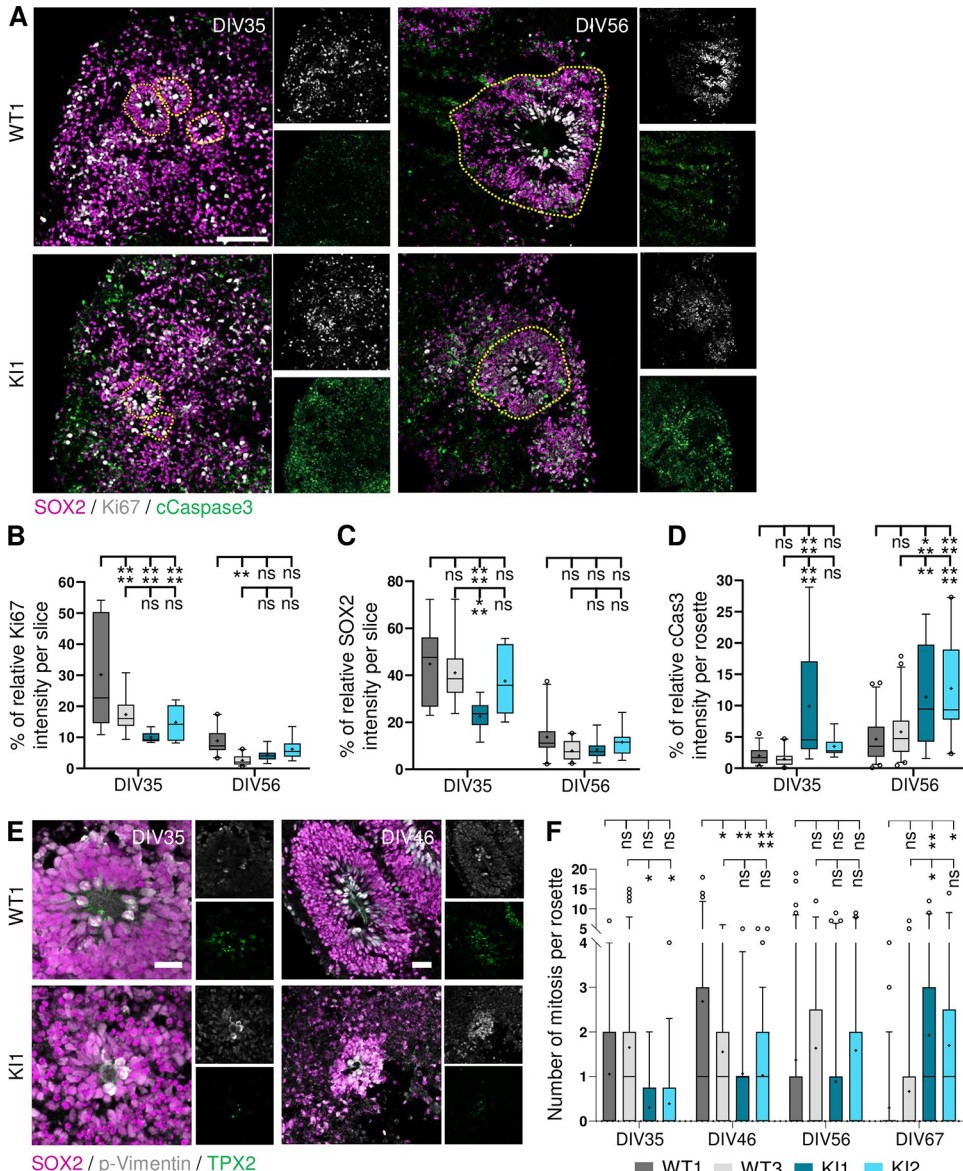

**Fig 7. *RTTN*-mutated organoids exhibit decreased proliferation and increased cell death at early stages.** All experiments were performed in wild-type (WT) and *RTTN*-mutated (KI) cortical organoids (CO). **(A)** Representative confocal images of proliferative (Ki67, grey) and apoptotic (cleaved Caspase-3, green) cells in CO at DIV35 and DIV56. SOX2 (magenta) labels neural progenitor cells. Neural rosettes are highlighted by yellow dashed lines. **(B-D)** Quantification of the percentage of Ki67+(B), SOX2+ (C) and cCaspase-3+ (D) cells such as seen in A, based on the relative intensity of the markers towards DAPI. Box-and-whisker plots display, in the box, the median (middle bar), the mean (cross) and the 25th-75th percentiles, and in whiskers, the 5th to 95th percentile of values from 4 different CO (n = 6 to 20 slices in total). **(E)** Representative confocal images of dividing cells (pVimentin, grey; TPX2, green) in WT and KI neural rosettes at DIV35 and DIV56. SOX2 (magenta) labels neural progenitors. **(F)** Quantification of mitotic events observed in neural rosettes such as seen in E. Box-and-whisker plots display, in the box, the median (middle bar), the mean (cross) and the 25th-75th percentiles, and in whiskers, the 5th to 95th percentile of values from 4 different CO (n = 6 to 20 slices total). ns, non-significant; *p-value<0.05; **p-value<0.01; ***p-value<0.001; ****p-value<0.0001 following two-way ANOVA with Tukey's correction (B-D, F). Scale bars: 100 μm (A), 50 μm (E).

pathogenicity by itself. Indeed, it rescues most of the tested *RTTN*-dKO-related phenotypes in RPE1 cells, and this as efficiently as the WT form (Figs 2 and S3), except for primary cilium formation (S4 Fig) and mitosis (Fig 2K). This result is somehow surprising since several

algorithms predicting pathogenicity of missense variants classify p.Arg985Gly as pathogenic (scores for Polyphen: 0.977; CADD: 29.6; AlphaMissense: 0.6), except REVEL that places it in an uncertainty range (score 0.37). This underscores the fact that even if prediction tools are becoming more and more efficient, they remain predictions; functional evaluation is the only way to provide trustworthy insight into variant pathogenicity. In addition to bringing information about the moderate damaging effect of the p.Arg985Gly variant *per se*, our study highlights here the crucial role of the 23 amino-acids (Ser963 to Arg985) that yet, do not encompass a known functional domain. These amino-acids are located outside the interacting domain with STIL (1–889 aa) that recruits Rotatin to centrioles [35]. Hence, we hypothesize that the deletion of the 23 amino-acids may alter its conformation and/or the interaction with Rotatin partners–other than STIL–that help it to be brought to the centrosome.

It is interesting to note that the levels of expression of the three isoforms do not vary much in the various cellular models used in this study (skin fibroblasts, iPSC, NSC and cortical organoids). Yet, we do not observe in each of these models all the cellular alterations we describe in total. Only slight alterations of cilium disassembly and ciliary function were observed in patient fibroblasts, while no defects of cell cycle and cell division were noticed (Figs 3 and S2). We could argue here that fibroblasts are differentiated cells that do not highly proliferate; nevertheless, our results contradict previous published work [29], which highlights the issue of individual/clonal variability. We found no significant alterations in iPSC (S6 Fig), which might be explained by their tolerance to variants and cellular abnormalities that preserves the stem cell physiology integrity [46]. Not surprisingly, given the severe brain malformations associated with *RTTN* variants, the most striking phenotypes were detected in the neural progenitor cells, cultured in 2D or 3D. It is of importance to note that clonal variability was observed in these models, highlighting the necessity to analyse several clones or to restore polyclonality by mixing several iPS clones. Despite some inconsistencies between clones, we brought out a general trend regarding cellular consequences downstream *RTTN* c.2953A>G. We demonstrated that *RTTN*-mutated NSC accumulated in G2-M phase due to numerous mitotic abnormalities that likely results in aneuploidy, cell arrest or cell apoptosis (Fig 4A–4H). These mitotic defects were associated to slight misorientation of the mitotic spindles that may favour a premature differentiation of NSC into neurons (Fig 4I and 4J). These alterations of NSC were also seen in cortical organoids (delayed proliferation/cell division, increased apoptosis) (Fig 7), and are thus likely to contribute to the reduction of the NSC pool, which is responsible for the overall reduced number and size of neural rosettes (Fig 6A–6C) and hence, of organoids (Fig 5), mimicking the microcephaly seen in the patient.

Furthermore, the use of cortical organoid model allowed us to discover a new role of Rotatin in the self-organization of NSC into neural rosettes. Indeed, we observed a reduced number of apical foci (positive for ZO-1, N-cadherin, and primary cilium) (Fig 6D–6F) in *RTTN*-mutated organoids, suggesting a defect in apico-basal polarity establishment of the bipolar NSC. Neural rosette formation is based on five different steps: cell intercalation, constriction, polarization, elongation and lumen formation [47]. Once formed, rosettes continue their expansion by fusing with each other [48]. The molecular mechanisms involved in these steps are not fully understood; they include various factors such as actin cytoskeleton remodeling, calcium, FGF2 or BMP signaling [47–50]. The question remains how the centrosome/primary cilium complex contributes to these steps.

Finally, it is noteworthy to remind here that we recently established a link between U4atac and primary cilium function [51]. Not only does it reinforce the possible feature overlap between *RNU4ATAC* and *RTTN* mutated patients, but it also stresses the involvement of primary cilium defects in the pathophysiology of primary microcephaly syndromes–even though this clinical feature is not among those of the well-known ciliopathies. Indeed, several studies

reported that alterations of cilium disassembly lead to a reduction of the proliferation rate of NSC [38–42], while another pointed out the role of cilium-related Shh pathway in NSC division into other progenitor cell types (basal radial glial cells, intermediate progenitors) necessary for brain size increase and folding [52]. Regarding *RTTN* deficiency, we believe that primary cilium defects (Figs 3 and S4) contribute, concomitantly to other major defects of cell cycle, to microcephaly. But, one could also argue that the ciliary defects are secondary to cell cycle abnormalities.

Altogether, through the study of the consequences of the only recurrent *RTTN* variant, we shed light on the pleiotropic functions of Rotatin, whose central role in brain development is starting to emerge.

## Materials and methods

### Ethics statement

Human primary fibroblasts were obtained from the CBC Biotec of the Hospices Civils de Lyon (certified with a specific French standard for biobanks, NF S96-900). Informed written consent for the use of the clinical data and/or biological samples in research was obtained from parents of the *RTTN* case and from those of the sex- and age-matched control. The study on patient cells was approved by the French national ethical committee Comité de Protection des Personnes (number 2021-A01551-40).

### Cell culture

Fibroblasts were cultivated in HAMF10 medium (Eurobio, CM1H1000-01), complemented with 12% fetal bovine serum (FBS, Eurobio, CVFSVF00-01) and 1% penicillin-streptomycin (PS, Gibco, 15140122). Human telomerase-immortalized retinal pigment epithelial cells (hTERT-RPE1) and derivatives, such as $RTTN^{+/+}$; $p53^{+/+}$ (control RPE1), $RTTN^{+/+}$; $p53^{-/-}$ (*p53*-KO), and $RTTN^{-/-}$; $p53^{-/-}$ (*RTTN*-dKO)-based RTTN-GFP-inducible cells (WT or variants) (S1 Methods) were grown in Dulbecco's modified Eagle DMEM/F12 medium, supplemented with 10% FBS. For inhibition of nonsense-mediated mRNA decay pathway, fibroblasts were treated with 100 μg/mL cycloheximide for 6 hours. For analysis of primary cilium, fibroblasts and RPE1-derived cell lines were maintained in culture medium with only 0.5% FBS for 48 hours; for cilium disassembly, FBS was reintroduced after 48-hour starvation for the time indicated in the figure. Induced Pluripotent Stem Cells (iPSC) issued from a healthy European male (PCi-CAU2) (Phenocell, Grasse, France) and its CRISPR/Cas9-edited derivatives (S1 Methods) were cultivated in 35-mm vitronectin-coated dish (STEMCELL Technologies, 7180) with mTeSR Plus medium (STEMCELL Technologies, 100–0276) supplemented with 0,1% PS. Cells were thawed in mTeSR Plus medium supplemented with 10 μM ROCK inhibitor Y-27632 (STEMCELL Technologies, 72302).

### iPSC-derived neural stem cell and neuron differentiation

The differentiation protocol was adapted from Boutaud *et al.* [53] with the following adjustments (S7A Fig). After 12 to 14 days in neural induction medium with medium replacement every other day, dense regions of iPSC appeared and were mechanically detached into small, squared clumps and transferred into a non-coated 35-mm dish with neural expansion medium (NEM) to form neurospheres overnight (S3 Table). The following day, neurospheres were plated onto Geltrex (Fisher Sci., 15180617) or poly-L-ornithine/laminin (PO/L)-coated dish (Sigma Aldrich, P3655/L2020) and medium changed every other day until neural rosettes appeared. Rosettes were then precisely excised and slightly crumbled before plating onto a

Geltrex or PO/L-coated dish (passage 0). From then on, neural stem cells (NSC) were cultivated in NEM and passed at high density (100,000 cells/cm$^2$) every 4 to 5 days using EDTA (Sigma Aldrich, E8008). From passage 3, cells were frozen in cryostore (STEMCELL Technologies, 07930) or cultivated until passage 10 maximum. To generate neurons, NSC were passed at low density (50,000 cells/cm$^2$) using EDTA and cultivated in Geltrex or PO/L-coated dish in N2B27 medium. Medium was changed every other day, and after 7 days in N2B27, 1 µL/mL laminin was added (Sigma Aldrich, L2020). NSC correct differentiation was evaluated through morphological changes (S7A Fig), downregulation of pluripotency genes and upregulation of neuroectodermal genes (S7B Fig) and expression of NSC markers by immunostaining (S7C Fig). For mitotic angles, NSC were blocked with 9 µM RO-3306 (Sigma, SML0569) for 18 hours. Upon release, 5 µM MG-132 (Sigma, M7449) was added to the media for 1 hour before fixation.

## Cortical organoid differentiation and growth curve

iPSC were cultivated in mTeSR+ on vitronectin-coated 35mm dishes until they reached 70% confluence. Using TrypLE Select (Fisher Sci., 11588846), iPSC were dissociated and 2,000 cells per well were plated in u-bottom ultra-low binding 96-well plates (ThermoFisher, 174925) with 150 µL of embryoid body medium (S4 Table). After seven days, medium was changed toward neural induction medium. At DIV21, organoids were put into differentiation medium containing B27 without vitamin A. At DIV31, B-27 was supplemented with vitamin A (Gibco, 11530536) (S9A Fig). Organoids were harvested at DIV35 for RNA extraction, and at DIV35, DIV46, DIV56 and DIV67 for cryosections and immunostaining. Growth curved was obtained by imaging twenty individual organoids, twice a week, and by measuring their area using Fiji software. Correct differentiation of CO was assessed by RT-qPCR analysis of gene expression of pluripotency and neuroectodermal markers at DIV35 (S9B Fig), and by immunostaining of neuroectodermal markers (PAX6, SOX2, Nestin) at DIV46 (S9C Fig).

## Flow cytometry analysis of cell cycle

Cells were plated at low confluence (20,000 to 50,000 cells/cm$^2$) and harvested with accutase two days later. After cells were washed with PBS, they were fixed for 1h at 4°C in 70% cold ethanol added drop by drop on a vortex. Cells were washed twice with PBS, and stained at room temperature for 30 minutes with 50 µg/mL propidium iodide (Sigma Aldrich, P4170) in presence of 100 µg/mL RNAse A (Sigma Aldrich, R6513). Flow cytometry analysis on 10,000 cells was performed using Canto II and results were analyzed with the FlowJo software v.10.9.0 (BD Biosciences).

## Immunofluorescence staining

Fibroblasts, RPE1 and NSC were respectively seeded on non-coated glass, Geltrex or PO/L-coated coverslips in 24-well plates at low density for cell cycle experiments, or at high density for cilium experiments in HAMF10, DMEM/F12 or neural expansion medium, respectively. When the desired confluence was obtained, medium was removed, and after a wash with PBS, cells were fixed in PBS-4% paraformaldehyde (PFA, EMS, 15713) for 20 minutes at room temperature (fibroblasts, NSC) or with ice-cold methanol at -20°C for 6 minutes (RPE1). Regarding organoids, they were washed in PBS and fixed in 4% PFA for 4 hours at room temperature with agitation. After two consecutive nights in 15% and in 30% sucrose, they were mounted in cryomolds with OCT embedding matrix and rapidly frozen with liquid nitrogen before being stored at -80°C. Cryosections of 14 µm-width were performed at -30°C with a Micron NX50 cryostat, and sections were conserved at -20°C on superfrost slides. Sections were allowed to

defrost 30 minutes at room temperature before being washed three times 10 minutes in PBS-0.5% Triton. Saturation was performed for 1 hour at room temperature with either, for monolayer cells, solution 1 (1X PBS, 10% normal goat serum (Merck Millipore, S26), 1% bovine serum albumin (BSA, Sigma, A9647), 0.1% Triton (Sigma, T8787)) or, for organoid sections, solution 2 (1X PBS, 2% BSA, 0.1% Tween20 (Sigma, P1379)). RPE1 cells were blocked with 3% BSA in PBS. Coverslips and sections were incubated overnight at 4˚C with primary antibodies (S5 Table) diluted in saturation solution, then washed three times with PBS and incubated with secondary antibodies in the saturation solution for 1h30 at room temperature in the dark. After they were washed thrice with PBS, they were incubated with 100 ng/mL DAPI (Invitrogen, D1306) for 10 minutes and washed. Coverslips were mounted on slides using FluorSave Mounting media (Sigma, 345789), and cryosections using Permafluor Aquaous Mounting Media (Fisher Sci., 12695925). Slides were imaged using the Zeiss LSM 800 confocal microscope (Carl Zeiss). All images were analyzed using the Fiji (ImageJ) software. For neural rosette measurements, the areas were quantified by detouring the neural rosette or the lumen. The thickness was assessed by the following formula: (mean of four rosette diameters–mean of four lumen diameters) divided by 2.

## Statistical analysis

All the data are reported as the mean with standard deviation (SD) or median with 95% confidence interval (CI) of at least three independent experiments. Normality of datasets was evaluated using the Shapiro-Wilk test, and outliers identified with the Grubb's test. All hypothesis tests were two-sided, and statistically significant differences (p-value<0.05) were calculated by parametric or non-parametric tests as indicated in figure legends. Statistical analyses were performed using GraphPad Prism software.

## Supporting information

**S1 Methods. Description of additional material and methods.**
(PDF)

**S1 Fig. Characterization of the impact of *RTTN* c.2953A>G variant on *RTTN* pre-mRNA splicing and global expression in patient fibroblasts.**
(PDF)

**S2 Fig. Characterization of the impact of *RTTN* c.2953A>G variant on cell cycle progression and mitosis in patient fibroblasts.**
(PDF)

**S3 Fig. Characterization of *RTTN*-dKO RPE1 cellular model and analysis of POC1B centriolar localisation.**
(PDF)

**S4 Fig. Characterization of the impact of p.Arg985Gly (RG) and Δ23 variants on primary cilium formation in *RTTN*-dKO RPE1 cellular model.**
(PDF)

**S5 Fig. Validation of the CRISPR/Cas9-mediated genetic modification of *RTTN* gene in four iPS clones.**
(PDF)

**S6 Fig. Characterization of *RTTN*-mutated iPS clone phenotypes.**
(PDF)

**S7 Fig. 2D differentiation of iPSC into neural stem cells (NSC) and neurons.**
(PDF)

**S8 Fig. Characterization of primary cilium formation and function in *RTTN*-mutated NSC.**
(PDF)

**S9 Fig. 3D differentiation of iPSC into cortical organoids.**
(PDF)

**S1 Table. Clinical features of all published cases with bi-allelic *RTTN* variants.**
(XLSX)

**S2 Table. Primer sequences.**
(PDF)

**S3 Table. Culture media for 2D neural differentiation protocol.**
(PDF)

**S4 Table. Culture media for cortical organoids.**
(PDF)

**S5 Table. Antibodies.**
(PDF)

## Acknowledgments

We thank the family and the patient for their contribution to this project, as well as Dr Ester Zuazo Zamalloa (Zumarraga Hospital, Gipuzkoa, Spain) for addressing the patient to us and transferring clinical data. We thank the Centre de Biotechnologie Cellulaire Biotec biobank for biosample management (Emilie Chopin, Isabelle Rouvet), Dr Gaetan Lesca for exome sequencing analysis, as well as Pr Caroline Schluth-Bolard and Dr Nicolas Chatron for the iPSC karyotypes. We also thank Isabelle Grosjean from the iPS_PGNM platform for her help and advice in cultivating induced pluripotent stem cells, Bruno Estebe (Imagine Institute, Paris) for his help in the establishment of the CRISPR-modified iPSC, and Leonardo Beccari for the useful insight into organoid cultures. We thank also all GenDev team members for constructive discussions as well as several intern students who helped to optimize experiments and acquire preliminary data: Lisa Malaisé, Grégoire Colomer, Aurora Devillaz, Lily-May Droulers, Noémie Gilibert, and Wassim Ouchetto. Finally, we thank the GenCiTy platform from the CRNL, and especially Sandrine Blondel who helped with the confocal settings for image acquisition, and Anne Ruiz who helped with the flow cytometry setting and acquisition.

## Author Contributions

**Conceptualization:** Ting-Yu Chen, Tang K. Tang, Sylvie Mazoyer, Marion Delous.

**Formal analysis:** Justine Guguin, Ting-Yu Chen, Alicia Besson, Eloïse Bertiaux, Nolan Ardito.

**Funding acquisition:** Virginie Hamel, Tang K. Tang, Sylvie Mazoyer, Marion Delous.

**Investigation:** Justine Guguin, Ting-Yu Chen, Alicia Besson, Eloïse Bertiaux, Audrey Putoux.

**Methodology:** Justine Guguin, Lucile Boutaud, Virginie Hamel, Sophie Thomas, Bertrand Pain, Tang K. Tang, Marion Delous.

**Project administration:** Marion Delous.

**Resources:** Miren Imaz Murguiondo, Sara Cabet, Patrick Edery, Audrey Putoux.

**Supervision:** Virginie Hamel, Sophie Thomas, Audrey Putoux, Tang K. Tang, Marion Delous.

**Validation:** Marion Delous.

**Visualization:** Justine Guguin, Ting-Yu Chen, Silvestre Cuinat, Eloïse Bertiaux, Sara Cabet, Marion Delous.

**Writing – original draft:** Justine Guguin, Ting-Yu Chen, Silvestre Cuinat, Sara Cabet, Marion Delous.

**Writing – review & editing:** Justine Guguin, Virginie Hamel, Sophie Thomas, Bertrand Pain, Patrick Edery, Audrey Putoux, Tang K. Tang, Sylvie Mazoyer, Marion Delous.

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
