## [Decision Letter · Decision Letter 0]

18 Sep 2024

Dear Dr. Delous,

Thank you very much for submitting your Research Article entitled 'A Taybi-Linder syndrome-related RTTN mutation impedes neural rosette formation in human cortical organoids' to PLOS Genetics.

The manuscript was fully evaluated at the editorial level and by independent peer reviewers. The reviewers appreciated the attention to an important topic but identified some concerns that we ask you address in a revised manuscript.

We therefore ask you to modify the manuscript according to the review recommendations. Your revisions should address the specific points made by each reviewer.

To resubmit, log into your Editorial Manager account and select the option 'Revise Submission' in the 'Submissions Needing Revision' folder.

Yours sincerely,

Gregory J. Pazour

Academic Editor

PLOS Genetics

Hua Tang

Section Editor

PLOS Genetics

As you can see in the comments below, both reviewers were positive about your manuscript but both had suggestions for improving the messaging. I don't feel that any additional experiments are needed, but please carefully address the concerns with text edits.

Reviewer's Responses to Questions

**Comments to the Authors:**

Reviewer #1: This is well written paper reporting a family with the clinical diagnosis of Taybi-Linder syndrome (TALS), a very rare monogenic disorder previously reported to result from RNU4ATAC mutations. The authors however identified a homozygous RTTN mutation instead that results in generation of 3 different aberrant transcripts as shown by detailled transcript analyses. Other mutations in RTTN have been previously identified in microcephaly, short stature and polymicrogyria with or without seizure” (MSSP, OMIM #614833). The identified allele has been reported in approx. 25% of MSSP families before so seems to represent a founder allele, however the effects of this mutation have not been studied before in detail.

The genetic findings in this paper a solid and certainly of interest to the human genetics and paediatrics/pediatric neurology community.

RTTN is a centrosomal protein and while previous studies suggest a role in procentriole elongation, cell cycle progression, mitosis and also cilia length regulation, it is not really understood how mutations exactly cause microcephaly. The authors have embarked on very detailled and laborious functional studies to investigate how their identified mutations could cause the CNS phenotype in their family, including use of patient fibroblasts, TP53-RTTN knockout RPE1 cells expressing wildtype and mutant RTTN in a doxycycline inducible manner expression and CRISPR/Cas9-edited pluripotent stem cells (iPSC) that they differentiated into neural stem cells (NSC) or cortical organoids (CO).

This functional work likewise is very solid and technically sound and involves a large amount of work, including months-long iPSC differentiation into cortical organoids. It proves pathogenicity of the identified RTTN mutation and I feel it’s a great strength of the paper that the authors have put in so much effort to study the effect of the human disease allele instead of using knockout models.

The identified mutation results in reduced levels of RTTN at centrioles and reduced centriole length as well as reduced levels of POC1B, a downstream centriole protein. The author further observe mild cell cycle progression defects upon expression of the mutant RTTN in RTTN Null cells. In contrast to a previous report, the authors don’t observe a cilia length defect in patient fibroblasts while RTTN knockout resulted in marked loss of cilia, likely the mutant RTTN retains enough function. They also see delayed cell cycle re-entry in patient fibroblasts. While no significant hedgehog pathway defect was observed, reduced ciliary ACIII was found in patient fibroblasts.

RTTN deficient iPSC cells showed cell cycle defects and increased cell death and slightly misaligned mitotic spindles. The authors suggest that a higher proportion of KI NSC could be prone to undergo asymmetric divisions and thus to prematurely shift to neurogenic cell divisions, which, as a result, would exhaust the progenitor pool. The authors then generated cortical organoids via neural differentiation of the iPSC (culture time 67 days), revealing delayed rosette formation, less-well organized rosettes, altered NSC polarization and reduced number of mitotic events when using RTTN mutant iPSC (carrying the human mutation).

Figure 1:

- Resolution in figure 1B appears low in my pdf, if this is a general issue and not down to the compression for the reviewer pdf, I would suggest a higher resolution here.

- I would suggest to add a pedigree of the family as well as a RTTN protein schematic showing the disease causing mutations and marking the associated phenotypic outcomes

- One could also move supp figure 1 a and B into figure 1, I feel it would be helpful for the readers to see this within the main manuscript rather than having to go to the supplement

Discussion

- Do you have a hypothesis why the patient has 11 pairs of ribs?

- “…hypothesize that misregulation of ciliary signaling, deriving from chemical or

mechanical cues, could alter NSC organization into neural rosettes in absence of functional

Rotatin.” Do you have a hypothesis why microcephaly is then not observed with other bona fide ciliopathies such

as BBS?

Supplementary clinical table:

- I would mention 11 pairs of ribs and consider an extra column displaying typical reported features of Taybi-Linder syndrome (TALS), or MOPD1 (OMIM #210710)

- There is a typo in the Corpus callosum ligne: “midly” insted of “mildly” in 2 instances

Reviewer #2: The authors dissect the mechanism of a recurrent RTTN variant in a rare autosomal syndrome characterized by microcephaly and other features. The identification of the recurrent variant c.2953A>G in their patient with clinical features of TALS does not have great news value, but does add to a general understanding of the phenotype associated with this RTTN variant. This is valuable given the rarity of this disorder (note: the authors provided a review on genotype/phenotype in all patients with this variant, which is added as supplemental data, and that is a nice addition to the report). The authors also performed functional cell-based assays and neural organoid assays to investigate the molecular mechanisms of disease associated with this variant, which is where the news value for PLOS Genetics readers really lies. The general mechanistic message (mitotic and spindle abnormalities, increased apoptosis, delayed neural rosette formation etc.) and the use of iPSC/organoid modeling is exciting, however, mathematical significance is not always consistent/clear in multiple Figures in this report, and discussion about these flaws are lacking. I feel the authors are somewhat overselling their data. In my opinion, this report certainly has potential, but requires major revision prior to acceptance in PLOS genetics. I do also want to note that this report was easy to read and data is well presented: thank you!

Major:

1. The authors do not comment at all on the differences between statistical significance/insignificance in iPSC/organoid findings between WT1/2/3 and KI1/2…. and iPSC/organoid data is where the news value of this manuscript lies. Some data are only significant when compared to WT1 but not WT2/3, and some data sets are significant only when comparing one KI line to WT controls, but not the other KI line that assesses the same variant. Examples in graphs in Fig 4J, 6C/D/F, 7C/F, S6G, S7H, S8B, S8C, S9G, S9F. There are definitely trends visible in (some of) the data, but inconsistencies in statistical significance in data also suggests that data should be interpreted with caution. Some findings are stronger than others. The authors need to address that in their manuscript results/discussion sections and rebuttal. Please, appropriately soften selected results and conclusions throughout the manuscript.

2. Is polarity initially established in rosettes, but then increasingly lost in KI1/2 clones? That is what it sort of looks like based on Figure 6A, but the quantification was only done in the DIV35 stage. It would be interesting to see quantification of polarity assessments of N-cadherin and ZO-1 in different divisional stages before and after DIV35. Also are cilia ever formed in these rosettes in WT populations and in KI clones? I am asking as you state in the discussion you propose that the problem may be ciliary function rather than formation, but did note that loss of RTTN can result in abnormal cilia formation/delayed disassembly. So, what do you see in organoids, which may better resemble the brain? Is there any cilia formation in WT and KI organoids that project into the centre of these rosettes (comparable to 3D cultures of renal cells that form a ball with a lumen inside wherein cilia project towards the centre)? If cilia form, are there any differences between WT and KI? I also wonder about this as cilia disassembly issues have been reported in Seckel syndrome (PMID: 35883578).

3. In the discussion of the manuscript the authors state that the RG isoform with p.Arg985Gly rescues as efficiently as WT RTTN. The RG isoform is definitely associated with a better rescue compared to delta 23, however, images such as Fig 2G/2K suggest it does have some impact on centriole length and spindle organization. Also, there is a slight decrease in number of ciliated cells and there seems to be a trend (albeit insignificant) of lower cilium length of +RG compared to +WT in RPE KO cells in Figures S4B and S4C. As such I don’t think that the rescue with +RG is as efficient as +WT. Also, note that not all in silico tools suggest that the amino acid change p.R985Gly is damaging, e.g. the REVEL score (0.37) is in the uncertain range. REVEL is a relatively new and potent prediction tool that I would say is used by most clinical labs these days. Can the authors please adjust their discussion/conclusions? Can they please add the REVEL score?

4. The discussion broadens to RNU4ATAC-associated microcephaly, but what else is known about microcephaly and centrosomes/primary cilia? It seems like there is more information in the literature. Please expand to a broader discussion of microcephaly in this respect (I found the following report in a quick search, but there should be more interesting literature out there: PMID: 35883578).

5. Please include references to specific Figures in your Discussion section of the manuscript when discussing your findings.

Minor:

1. Please refrain from using the word ‘mutation’. The use of ‘variant’ at least in the context of human genetics is more appropriate. Please, replace the word mutation for variant in the title, abstract and throughout the remainder of the manuscript.

2. Abstract: p.3, line 30: The Taybi-Linder syndrome (TALS) is a rare genetic disorder >>> Taybi-Linder syndrome (TALS) is a rare autosomal recessive disorder…

3. Author summary (p.4, line 55). “…and carries a homozygous mutation, the only recurrent one, in the centrosomal gene RTTN” > “and is homozygous for a recurrent, pathogenic variant in the RTTN gene that encodes a centrosomal protein.” Note: please refrain from using the word “carry” when it does not relate to carrier testing (i.e. when parents are heterozygous for the variant). There are multiple instances where ‘carry’ is used erroneously. Please correct throughout the manuscript.

4. Introduction: p.5, line 76: please, add variant in HGVS nomenclature (https://hgvs-nomenclature.org/stable/) and include RefSeq (i.e. NM_xxxx(RTTN):c. and p.). I realize that you did add this information in the Supplementary Table (thank you!), but it is important to also have this information in the main manuscript so scientists do not have to dig deep to find it. Please also specify what type of variant this is here.

5. You use the word link/linked frequently. Where possible, please, explain. For example, p.6, line 88: “Alterations of all these processes have been linked to microcephaly.” >>> Alterations of all these processes are thought to contribute to the development of microcephaly.

6. Results. P7, line 100: Please define what sample you received.

7. Results. P7, line 104: I am not a clinician, but think it is may be more appropriate to provide percentiles for weight/height. What does “-DS” mean? Standard deviations? There are more growth parameters further down in this paragraph. Please also address those.

8. Results. P7, line 122. Please add: This variant was reported in the ClinVar database as pathogenic (Variation ID: 977819).

9. Figure S7E: please add when cells were harvested in the legend.

10. Figure S1C: What is the arch with 4? Is that a transcript not described? Could there be more transcripts that the three investigated (not visible with RT-PCR due to quantity or resolution)? I have no experience with RNAseq and am not that familial with these plots. Please, explain.

11. You found slight different abnormalities in fibroblasts, iPSC, NSC. Are there examples of other studies where cellular defects of fibroblasts, iPSC and organoids are slightly different? If you cannot compare to cortical organoids, are there other types of organoids with slight differences compared to iPSC and other cell types in literature? I am not criticizing data here, but just wondered if others have reported discrepancies in other studies (unrelated to RTTN).

12. Page 9, line 161: “In patient cells, we observed a reduced quantity of Rotatin…” > “In patient cells, we observed a modest reduction in quantity of Rotatin…”

13. Page 13, line 267: you claim that you see the same proportions of all three isoforms, but the proportion of RG seems lower to me in S9E.

Textual:

1. Abstract: p3., line 32: “It is caused by mutations in RNUATAC whose transcript, the small nuclear RNA U4atac,is involved in the excision of ~850 minor introns” > “It is caused by pathogenic variants in the RNUATAC. Its transcript, the small nuclear U4atac, is involved in the excision of ~850 introns”.

2. Abstract: p.3, line34: “…but no mutation in RNU4ATAC, instead the RTN c.2953A>G variant at the homozygous state.” This sentence seems incomplete. Suggest to change to “…but no pathogenic variant was found in RNU4ATAC, instead a homozygous RTTN c.2953A>G variant was detected by whole-exome sequencing”.

3. Introduction: p.5, line 69: Since the identification in 2011 of RNU4ATAC, … >>> since the identification of RNU4ATAC in 2011, …

4. Introduction:p5, line 74: Here we report on a particular case of a patient with TALS traits but no variant in RNU4ATAC > Here we report on a patient with clinical features of TALS but no variant in RNU4ATAC.

5. Introduction: p.5line 75: carries > has

6. Introduction: p.78… 37 patient with bi-allelic pathogenic variants in RTTN have been reported >>> 37 patients with bi-allelic pathogenic variants in RTTN have been reported (# of these patients from # families had the c.2953A>G variant).

7. Introduction: p.5, line 80: slopping forehead > sloping forehead

8. Introduction: p.5, line 82: adult-hood > adulthood

9. Results: p.7, line 99: TALS-like traits > TALS-like features

10. Results:p.7, line109: “stature weight growth retardation”. I am not sure if such a word exists. Do you mean growth retardation with respect to both weight and height?

11. Results: p.7, line 115: microenencephaly. Did you mean microcephaly?

12. Results: p.7, line 120: Then, a whole exome sequencing was done > Then, whole exome sequencing was done

13. Results:p.7, line 121: add RefSeq (NM_xxx) for RTTN

14. Results: p8, line 125: “a boy and a girl, also homozygous…” > “a boy and a girl, are also homozygous…” I am not sure what ‘although not described as such’ refers to in this sentence. Do you mean the authors speak of MOPD1, but not TALS? You did not really explain it clearly, but is TALS a comprehensive term for both MOPDI and MOPDIII?

15. Results: p.8, line 133: present quite heterogeneous > present with quite heterogeneous

16. Results: p8, line 137: we undertook to investigate c.2953A>G further through the study of various cellular models, which will also give us insights for the Taybi-Linder syndrome >>> we investigated c.2953A>G further through the study of various cellular models, which provided us with insights in the molecular mechanisms of disease associated with Taybi-Linder syndrome.

17. Page 12, line 235: please add at what stage NSC cells were harvested.

18. S9H in the lower graph is mislabeled as “F” in the Figure S9. The legend does correctly refer to H.

**Have all data underlying the figures and results presented in the manuscript been provided?**

Reviewer #1: Yes

Reviewer #2: Yes

PLOS authors have the option to publish the peer review history of their article (what does this mean?). If published, this will include your full peer review and any attached files.

Reviewer #1: No

Reviewer #2: No

---

## [Decision Letter · Decision Letter 1]

13 Nov 2024

PGENETICS-D-24-00844R1A Taybi-Linder syndrome-related RTTN variant impedes neural rosette formation in human cortical organoidsPLOS Genetics Dear Dr. Delous, Thank you for submitting your manuscript to PLOS Genetics. One reviewer still has a concern about the connection between cilia and microcephaly.  It seems that this is an important point that should be considered.  Please either edit the manuscript to address the point or explain why this is not relevant. Please submit your revised manuscript within 30 days Dec 13 2024 11:59PM. If you will need more time than this to complete your revisions, please reply to this message or contact the journal office at plosgenetics@plos.org. Please include the following items when submitting your revised manuscript:*
A rebuttal letter that responds to each point raised by the editor and reviewer(s). You should upload this letter as a separate file labeled 'Response to Reviewers'. This file does not need to include responses to formatting updates and technical items listed in the 'Journal Requirements' section below.*
A marked-up copy of your manuscript that highlights changes made to the original version. You should upload this as a separate file labeled 'Revised Manuscript with Track Changes'.*
An unmarked version of your revised paper without tracked changes. You should upload this as a separate file labeled 'Manuscript'. If you would like to make changes to your financial disclosure, competing interests statement, or data availability statement, please make these updates within the submission form at the time of resubmission. Guidelines for resubmitting your figure files are available below the reviewer comments at the end of this letter. We look forward to receiving your revised manuscript. Kind regards, Gregory J. PazourAcademic EditorPLOS Genetics Hua TangSection EditorPLOS Genetics Aimée DudleyEditor-in-ChiefPLOS Genetics Anne GorielyEditor-in-ChiefPLOS Genetics **Additional Editor Comments (if provided):**    **Journal Requirements:****Reviewers' comments:** Reviewer's Responses to Questions

**Comments to the Authors:**

Reviewer #1: I feel overall the points raised have been addressed sufficiently.

Both reviewers raised the point to broaden the discussion of potential microcephaly-cilia-relationships. The authors themselves now state in the discussion page 19/20 that microcephaly is not a clinical features of classic ciliopahies (I agree), my only remaining minor point would be to maybe very briefly discuss here the possibility that ciliary defects or delayed deciliation observed in conjunction with reduced NPC proliferation and subsequent microcephaly may be a secondary effect of cell cycle progression issues, causing both delayed deciliation and reduced proliferation, rather than primarily delayed deciliation causing cell cycle progress issues (this is kind of the hen and the egg question...).

If deciliation/cilia length issues are secondary in microcephaly disorders, this could explain the lack of microcephaly phenotypes in primary ciliary diseases.

**Have all data underlying the figures and results presented in the manuscript been provided?**

Reviewer #1: Yes

PLOS authors have the option to publish the peer review history of their article (what does this mean?). If published, this will include your full peer review and any attached files.

Reviewer #1: No

 **Figure resubmission:** While revising your submission, please upload your figure files to the Preflight Analysis and Conversion Engine (PACE) digital diagnostic tool, https://pacev2.apexcovantage.com/. PACE helps ensure that figures meet PLOS requirements. To use PACE, you must first register as a user. Registration is free. Then, login and navigate to the UPLOAD tab, where you will find detailed instructions on how to use the tool. If you encounter any issues or have any questions when using PACE, please email PLOS at figures@plos.org. Please note that Supporting Information files do not need this step. If there are other versions of figure files still present in your submission file inventory at resubmission, please replace them with the PACE-processed versions. **Reproducibility:** To enhance the reproducibility of your results, we recommend that authors deposit laboratory protocols in protocols.io, where a protocol can be assigned its own identifier (DOI) such that it can be cited independently in the future. Additionally, PLOS ONE offers an option to publish peer-reviewed clinical study protocols. Read more information on sharing protocols at https://plos.org/protocols?utm_medium=editorial-email&utm_source=authorletters&utm_campaign=protocols

---

## [Editor Report · Decision Letter 2]

27 Nov 2024

Dear Dr Delous,

We are pleased to inform you that your manuscript entitled "A Taybi-Linder syndrome-related RTTN variant impedes neural rosette formation in human cortical organoids" has been editorially accepted for publication in PLOS Genetics. Congratulations!

Yours sincerely,

Gregory J. Pazour

Academic Editor

PLOS Genetics

Hua Tang

Section Editor

PLOS Genetics

Aimée Dudley

Editor-in-Chief

PLOS Genetics

Anne Goriely

Editor-in-Chief

PLOS Genetics

Comments from the reviewers (if applicable):

**Data Deposition**

http://datadryad.org/submit?journalID=pgenetics&manu=PGENETICS-D-24-00844R2

**Press Queries**

---

## [Editor Report · Acceptance letter]

10 Dec 2024

PGENETICS-D-24-00844R2 

A Taybi-Linder syndrome-related RTTN variant impedes neural rosette formation in human cortical organoids 

Dear Dr Delous, 

We are pleased to inform you that your manuscript entitled "A Taybi-Linder syndrome-related RTTN variant impedes neural rosette formation in human cortical organoids" has been formally accepted for publication in PLOS Genetics! Your manuscript is now with our production department and you will be notified of the publication date in due course.

With kind regards,

Anita Estes

PLOS Genetics

On behalf of:
